# Design of and Experiments with an Automatic Cuttage Device for an Arch Shed Pillar with Force Feedback

**Kezhou Chen, Xing Liu, Shiteng Jin, Longfei Li, Xin He, Tao Wang, Guopeng Mi, Yinggang Shi** 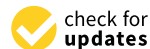 **and Wei Li \***

College of Mechanical and Electronic Engineering, Northwest A&F University, Xianyang 712100, China; ckz@nwafu.edu.cn (K.C.); njhlxhn@nwsuaf.edu.cn (X.L.); jin2020010193@nwafu.edu.cn (S.J.); llf@nwafu.edu.cn (L.L.); xnhexin@nwafu.edu.cn (X.H.); wangtao666@nwafu.edu.cn (T.W.); miguopeng@nwafu.edu.cn (G.M.); syg9696@nwsuaf.edu.cn (Y.S.)
\* Correspondence: liweizibo@nwsuaf.edu.cn; Tel.: +86-139-0925-8177

**Abstract:** In order to realize the automatic cutting of arch shed pillars, an automatic cuttage device for an arch shed pillar with force feedback was designed in this study. First, the wind resistance of the arch shed was simulated and analyzed using ANSYS, and the cuttage depth of the arch shed pillar was determined. According to the environment for the cuttage operation of the arch shed pillar and the agronomic requirements, such as the arch shed span, arch shed height, and cuttage depth, the function, structure, and basic design parameters of the arch shed automatic cuttage device were determined. Then, to reduce the damage rate of the pillar and achieve equal-depth cuttage, a force feedback system for the actuator of the cuttage device was constructed to estimate the cuttage resistance and depth in real time. To reduce the impact of the starting and stopping of each motor in the actuator, trajectory planning of the execution end in the pillar transfer stage was performed in the Cartesian coordinate system. The motion law of portal trajectory based on the Láme curve was analyzed, and MATLAB simulations were used to solve the relevant motion parameters. In addition, the modality of key components of the cuttage device was simulated and analyzed by using the SOLIDWORKS simulation plug-in. Finally, the experimental prototype was constructed according to the simulation results. The simulation and field cuttage experiments showed that the cuttage device produced equal-depth cuttage for the arch shed pillar, where the depth of the arch shed pillar was 10 cm, the average cuttage time of a single pillar was 6.2 s, and the error of the cuttage depth was ±0.5 cm in wet soil. The operation of the device was stable, as evidenced by the smooth and mutation-free operation trajectory and speed curve of the execution end. The results of the modal experiment suggest that resonance would not occur during the operation for resonance frequencies between 303 Hz and 565 Hz. This arch shed pillar automatic cuttage device has an optimal operation performance and meets the agronomic requirements of arch shed pillar cuttage.

**Keywords:** arch shed; automatic cuttage; force feedback; simulation design; trajectory planning

## 1. Introduction

Mulching film cultivation technology in arch sheds can preserve heat, moisture, and soil, improving the survival rate and shortening the planting cycle of crops [1–3]. At present, the embedded depth of an arch shed pillar depends on the experience of farmers, which can lead to poor wind resistance of the arch shed and is not conducive to the automatic construction of arch sheds. Many literature studies exist on automatic cuttage devices for arch sheds, such as the arch shed cuttage and filming machine developed by Fujiki Agricultural Machinery Manufacturing Institute (Higashiosaka, Japan), the arch shed frame construction machine developed by French CM-REGERO Industries (La Chapelle-Basse-Mer, France) and RABAUD (La Chapelle-Basse-Mer, France), the arch shed cuttage machine developed by Dubios Agrinovation Inc. (Saint-Rémi, Canada ), the arch shed cuttage machine jointly developed by Mark William, Tunnelmatic Inc. (RABAUD La



Chapelle-Basse-Mer, France), and LesAgrisudvende Inc. (Libourne, France), and the arch shed automatic cuttage device developed by Liu et al. [4,5]. In China, the planting area that utilizes arch shed film mulching cultivation technology reaches 13,000 square kilometers every year [6,7], which requires many automatic cuttage machines. However, the sizes of the currently studied arch shed cuttage machines and cuttage devices are large, which is insufficient to meet the requirements of the small-scale operating environment of the Chinese farmland. These devices require auxiliary manual operation with less automation. The pillar can also be easily broken during cuttage in a hard-soil environment. In order to solve these problems, we aimed to develop an automatic cuttage device for arch shed pillars with a pillar protection function that is suitable for small field operations.

According to previous studies, the small arch sheds used for seedling cultivation in Shaanxi and Gansu have a span of 60 cm and a height of more than 35 cm (Figure 1). The most commonly used arch shed pillar has a span of 60 cm, a height of 50 cm, and a diameter of 1 cm, with a spacing of 1 m between adjacent pillars. To ensure wind resistance of the arch shed, "equal-depth cuttage" should be adopted for the pillar. At the same time, it is necessary to avoid film rupturing caused by pillar damage, which can lead to a decline in the heat preservation performance of arch sheds [8,9]. During the automated cuttage operation of the arch shed, it is important to first achieve automatic cuttage with a lower damage rate, and then ensure the consistency of the highest point of the 1 cm arch shed after the cuttage at the same time—namely, equal-depth pillar cuttage.

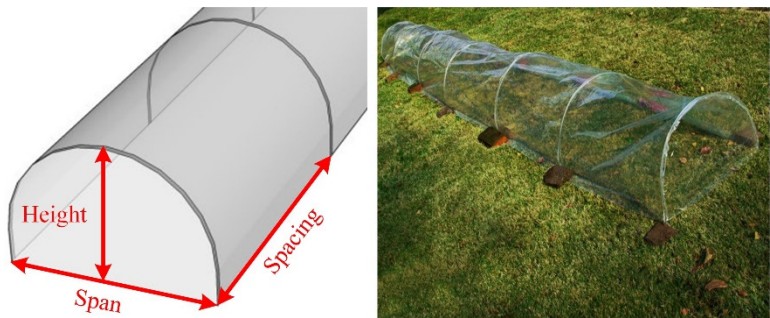

**Figure 1.** Plastic-film-covered cultivation in an arch shed.

According to the operation environment and the experience of farmers, the basic requirements for the automatic cuttage of arch sheds are as follows:

(1) The cuttage span of the arch shed pillar should be 60 cm with less than 1 cm of error;
(2) The height of the arch shed pillar after cuttage should be 40 cm with less than 1 cm of error;
(3) The diameter of the arch shed pillar that can be cut should be 1 cm;
(4) The cuttage depth should be equal at the ends of the arch shed pillar;
(5) The damage rate of the arch shed pillar should remain below 5%;
(6) The cuttage device should operate smoothly, have optimal stability, and have a long service life.

According to agronomic requirements and to the operation environment of arch shed pillars, the process of the automatic cuttage of arch shed pillars was constructed, as shown in Figure 2. In this process, the cuttage device installed on the chassis moved forward along a straight line, stopped after reaching the cuttage position, and performed a pillar cuttage.

In the automatic arch shed pillar cuttage process, the span and height can be determined through the mechanical design of the cuttage device, and the diameter of the arch shed pillar can be customized in advance. However, several problems still need to be addressed:

(1) Suitable pillar cuttage depth;
(2) A quick and convenient method for determining the ideal cuttage depth at both ends of the pillar;

(3)     Ensuring that the damage rate of the pillar is controlled below 5%;
(4)     Ensuring the optimal operation and stability of the device.

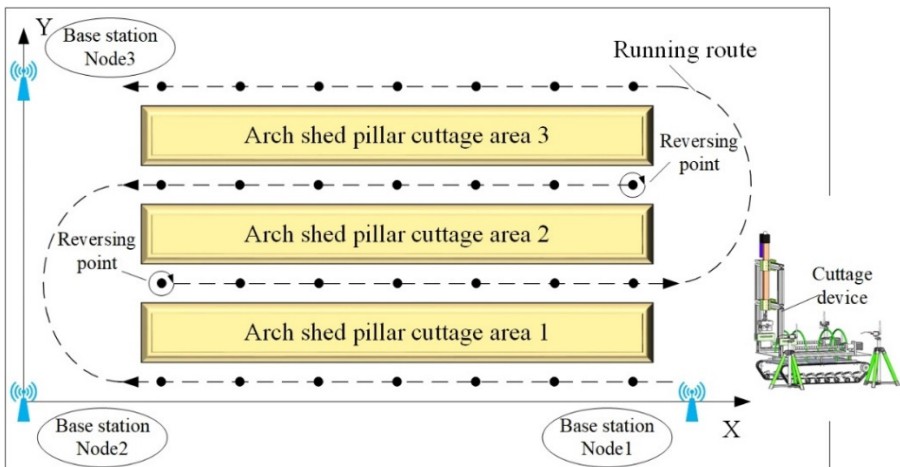

**Figure 2.** Automatic cutting process of arch shed pillars.

Wind resistance is an important index for evaluating the firmness of a small arch shed [10]. The pillar cuttage depth determines the wind resistance of the arch shed. The analyses of wind resistance for greenhouses have mostly focused on the environment of the greenhouse [11–13]. A small arch shed covered with a flexible plastic film is strongly sensitive to wind load. The wind resistance of an arch shed is related to the spacing, span, and embedded depth of the arch shed pillar [14]. However, the current embedded depth of arch shed pillars is essentially determined based on experience, which is not supported by the theoretical data.

According to the agronomic requirements for the cuttage of arch shed pillars in a greenhouse, soft soil with a rotary tillage depth of about 25 cm is suitable during cuttage, while the embedded depth of the arch shed pillars should be greater than 5 cm. The stability of an arch shed increases with an increase in cuttage depth. However, since the size of an arch shed pillar is fixed, increasing the cuttage depth inevitably reduces the height of the arch shed and weakens the preservation of heat and moisture. The mechanical characteristics of the pillar at different cuttage depths and under different levels of wind force can be determined using a statistical analysis of the wind force during the spring in Shaanxi, China. In other words, a reasonable pillar cuttage depth that meets the wind resistance of the arch shed can be deduced by analyzing the wind resistance of the arch shed pillar at different cuttage depths.

To achieve equal-depth cuttage of the arch shed pillar, it is necessary to design a detection mechanism for a stable cuttage depth to monitor the embedded depth of the arch shed pillar in real time. However, this task is currently difficult to implement. Mossadeghi-Björklund et al. [15] reported a positive correlation between soil penetration resistance and penetration depth. Based on this principle, monitoring of the cuttage resistance can be used to detect the cuttage depth.

Therefore, the development of real-time monitoring for cuttage resistance has become the focus and difficulty of research. Many relevant studies have been conducted in different research fields, including industries and services concerning torque detection. According to the law that the resistance moment of a joint will increase rapidly and the current of a joint motor will jump when the manipulator is collapsed, Spong et al. [16] constructed a flexible joint that could estimate the impact moment value of the manipulator joint by using the current jumping value of the joint motor. Chen et al. [17] proposed a general sensorless collision-detection algorithm for robot actuator faults. Shi et al. [18] designed a flexible picking manipulator with an obstacle-avoidance function that could detect the collision resistance of the manipulator during fruit and vegetable picking. For the monitoring of

cuttage depth, a cuttage depth feedback mechanism based on "current–force–depth" was constructed in this study, and a cuttage resistance feedback device based on an electric push rod was built. A cuttage-resistance feedback algorithm was designed that could be used to explore the relationship between the current jumping value and the cuttage resistance. Moreover, the resistance of the electric push rod was obtained by detecting the motor current, and the cuttage depth of the pillar was indirectly obtained according to the relationship between the resistance and cuttage depth.

Excessive impact force leads to the damage of the pillar cuttage. Impact force is a result of hard obstacles under the soil layer that are encountered during the pillar cuttage. The hard obstacles under the soil layer are uncontrollable factors, which can be avoided by the sensing ability of the force through the execution end based on the detection of cuttage resistance using the "current–force–depth" feedback system.

In the process of transporting the pillar for cuttage, the curves of the operation trajectory, speed, and acceleration at the execution end are not smooth, which makes the cuttage device subject to impact force. Therefore, a trajectory curve was planned to ensure smooth curves of the operation trajectory, speed, and acceleration. Meanwhile, the vibration characteristics of the cuttage device were analyzed to obtain it's inherent frequencies, which was used to prevent resonance, effectively reduce the impact force, and ensure the optimal operation of the device.

To solve the above-mentioned basic requirements for the automatic cuttage of arch shed pillars, the wind resistance of the arch shed was analyzed in this study and the cuttage depth of the arch shed pillar was determined. A real-time cuttage depth detection mechanism based on "current–force–depth" was constructed for the real-time monitoring of the cuttage depth. According to basic parameters, including span, height, cuttage depth, and pillar diameter, the gantry-shaped cuttage device was simulated and designed to analyze the motion trajectory, speed, and acceleration curves at the execution end, and the rationality of size parameter selection and the smoothness of the motion trajectory were verified. A modal analysis of the key components of the automatic cuttage device was performed to determine the first four order modal frequencies of the cuttage device, which prevented the occurrence of resonance, eliminated impact force, and ensured the smoother operation of the device. Finally, the prototype machine was manufactured and the cuttage experiment was performed.

## 2. Pillar Cuttage Depth and Wind Resistance Analysis of the Arch Shed

Wind blowing vertically against the side of the arch shed pillar has the greatest impact on the arch shed. ANSYS was used to analyze the wind resistance of the arch shed at different cuttage depths. The vertical wind on the side of the arch shed is shown in Figure 3. From the above information, it was calculated that the height of the arch shed poles should be $H = 50$ cm and the span should be $b = 60$ cm. The height of the arch shed is h and the embedded depth of the arch shed pillar is $a = H - h$ Qin et al. [4,19,20] found that the appropriate height-to-span ratio for an arch shed is $h/b = 0.5 \sim 0.67$, where $h = 35 \sim 40$ cm and $a = 10 - 15$ cm, which can be obtained from the height requirements of the arch shed.

In Yangling, Shaanxi, China, the values for altitude and air density are $z = 435 \sim 563$ m and $\rho = 0.00125e^{-0.0001z}$, respectively, and the average ground wind speed from March to April is level 2–3, with occasional strong winds of level 7–8 [21–23]. To ensure safety during operation, the pressure of a level 10 wind acting on the arch shed was considered to be the wind pressure limit; at this time, the basic wind pressure is $\omega_0 = \rho v^2/2 = 420$ N/m$^2$ [24,25]. The arch shed was a non-high-rise building with a height of less than 30 m. When the wind vibration coefficient is $\beta_z = 1$, the variation coefficient of the wind load height is $\mu_z = 1$, and the wind load shape coefficient is $\mu_s = 0.2$ [26,27], then the wind load applied to the arched shed is $\omega_k = \omega_o \beta_z \mu_z \mu_s = 84$ N/m$^2$.

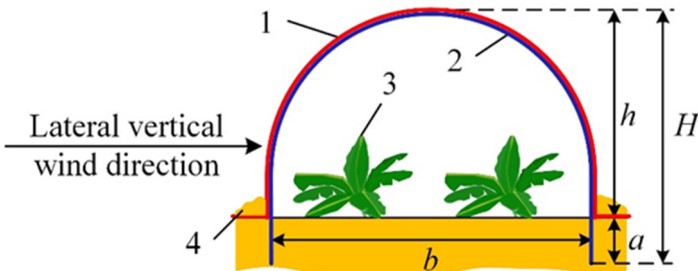

**Figure 3.** Schematic diagram of vertical wind on the side of arch shed. 1: plastic film; 2: arch shed pillar; 3: plant; and 4: soil. Note that b: span of arch shed pillar; H: height of arch shed pillar; h: height of arch shed and a: the embedded depth of the arch shed pillar.

Under an external temperature of 5 °C–10 °C, the compressive yield strength of the pillar material is $R_m = 2.5 \times 10^8$ Pa, the elastic modulus is $E = 1.67 \times 10^{11}$ Pa, $G = 7.69 \times 10^{10}$ Pa, $\mu = 0.3$, Young's modulus is $2 \times 10^{11}$ Pa, and the density is $\rho = 7850$ kg/m$^3$. The arch shed pillar model was established using SOLIDWORKS and imported into ANSYS Workbench for a simulation analysis of stress, strain, and deformation to evaluate the wind resistance of the arch shed pillar under a level 10 wind. Tetrahedronal elements were used to divide the grid, and the minimum grid unit size was set to 0.5 cm. Under the same wind load, fixed constraints at both ends of the arch shed pillar were added to simulate the fixation of the pillar by soil after the arch shed pillar was cut into the soil. A wind load of 84 N/m$^2$ was applied on one side of the arch shed to simulate and analyze the stress of the arch shed pillar under a level 10 wind. When the cuttage depth of the pillar ranged from 8–15 cm, the stress and strain of the arch shed pillar increased with each 1 cm-increase in the cuttage depth (Figure 4). The simulation results revealed that the stress and strain of the arch shed pillar decreased with an increase in the pillar's embedded depth from 8 cm to 15 cm, and it was more resistant to wind and snow disasters. When the cuttage depth was greater than 10 cm, the downward trend of stress and strain became less pronounced.

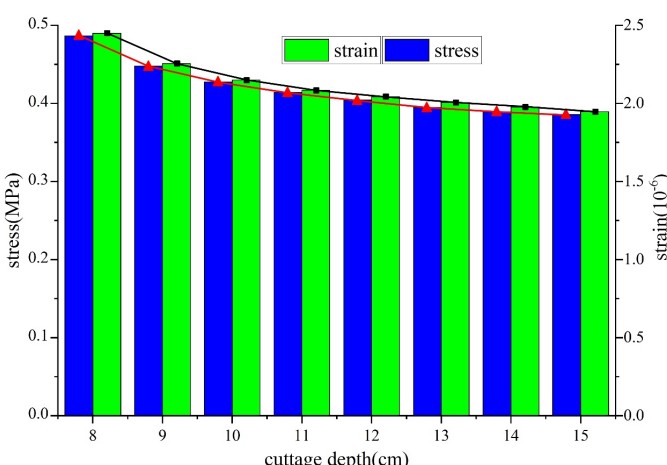

**Figure 4.** Pillar cuttage depth and wind resistance analysis of the arch shed.

When the height of the small arch shed increased, the living space of the plant increased as well, which was more conducive to the growth of seedlings. Therefore, within the appropriate height-to-span ratio of 0.5–0.67 cm and with $h = 40$ cm, the embedded depth of the arch shed pillar would be $a = H - h = 10$ cm. Under these parameters, the strain of the arch shed pillar was $2.15 \times 10^{-6}$, which was considerably lower than the allowable strain of the structural steel, namely $[\varepsilon] = 3.74 \times 10^{-4}$, and the equivalent stress value was 0.427 MPa, which was also considerably lower than the allowable stress of

the structural steel, namely $[\sigma] = 62.5$ MPa. Therefore, the arch shed pillar was expected to withstand a level 10 wind.

### 3. Real-Time Detection Principle of Cuttage Depth Based on "Current–Force–Depth"

Assuming the use of a DC motor, the current of the motor would increase rapidly with an increasing load [28,29]. Based on this observation, an electric push rod was used to cut the arch shed pillar, and a stress analysis diagram of the pillar cuttage using an electric push rod was constructed, as shown in Figure 5. During the cuttage operation of the electric push rod, the resistance of the electric push rod would increase rapidly and the current driving the motor would correspondingly increase with an increase in the embedded depth of the pillar. In other words, as the cuttage depth of the pillar increased, the cuttage resistance increased, the resistance of the electric push rod also changed, and the current driving the motor would also increase. According to this principle, the electric push rod caused the brushless motor (1) to rotate the ball screw (4) after being decelerated by the secondary reducer (2). The rotation of the ball screw (4) caused the nut assembly (3) that is fixedly connected with the push pole (5) to move, thus converting the rotating motion of the brushless motor (1) into a linear motion of the push rod (5) and generating axial thrust. Subsequently, the relationship between the current change of the motor and the electric push rod, as well as the embedded depth of the arch shed pillar, can be calculated to detect the motor current and to infer the torque changes with the depth during cuttage.

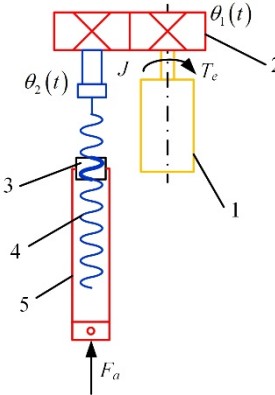

**Figure 5.** Stress analysis diagram of the pillar cuttage using an electric push rod. 1: brushless motor; 2: reducer; 3: nut assembly; 4: ball screw; and 5: push pole.

According to the literature, the external force on the electric push rod during operation includes the driving torque of the motor and the cuttage resistance. Let the equivalent moment of inertia on the motor spindle be $J$, the equivalent system damping on the motor spindle be $B$, the motor angular displacement be $\theta_1(t)$, the lead screw angular displacement be $\theta_2(t)$, the electromagnetic torque be $T_e$, and the load torque be $T_L$. Then, the torque balance equation on the motor shaft is:

$$T_e = J\ddot{\theta}_1(t) + B\dot{\theta}_1(t) + T_L \tag{1}$$

Let the lead screw be $p_h$, the transmission efficiency of the ball screw pair be $\eta$, and the cuttage resistance be $F_a$. Then, the relationship between cuttage resistance and the load torque is as follows:

$$T_L = \frac{F_a p_h}{2\pi\eta} \tag{2}$$

Let the current of the motor in a steady state be $i$, the electromagnetic torque be $T_e$, and the motor torque coefficient be $K_t$. Then, the electromagnetic torque equation is:

$$T_e = K_t i \tag{3}$$

Combining Equations (1)–(3) yields the following relationship between the motor current and the cuttage resistance:

$$i = \frac{F_a p_h}{2\pi\eta} + \frac{J\ddot{\theta}_1(t)}{K_t} + \frac{B\dot{\theta}_1(t)}{K_t} \tag{4}$$

According to Equation (4), when the cuttage device works normally, the current value flowing through the electric push rod is related to the cuttage resistance of the arch shed. Therefore, the cuttage resistance can be estimated by detecting the current value of the motor.

As mentioned above, the cuttage resistance can be obtained by monitoring the motor's current value. Therefore, the mathematical model of pillar cuttage resistance and cuttage depth needs to be established. According to the literature, when the dimensionless parameters related to the soil type, $C_1, C_2, C_3$ and $C_4$ are adopted, the area of the conical cross-section of the front end of the arch shed pillar is S, the unit of soil bulk density is $D_b$, the weight of the soil moisture is $\theta_g$, and the penetrating depth of the arch shed pillar is $h$ [30], the relationship between the cuttage resistance and the penetrating depth is as follows:

$$F_a = S \times C_1 D_b{}^{C_2}\theta_g{}^{C_3}h^{C_4} \tag{5}$$

Consequently, the relationship between the current of the electric push rod driving the motor and the penetrating depth is:

$$i = \frac{S \cdot C_1 D_b{}^{C_2}\theta_g{}^{C_3}h^{C_4} p_h}{2\pi\eta} + \frac{J\ddot{\theta}_1(t)}{K_t} + \frac{B\dot{\theta}_1(t)}{K_t} \tag{6}$$

According to Equation (6), when the electric push rod is operational, the motor current is linearly related to the penetrating depth. Therefore, the detection algorithm for the cuttage depth of the arch shed pillar can be constructed based on the current feedback by analyzing the equation for the relationship between the motor current of the electric push rod and the cuttage depth of the arch shed pillar. The cuttage depth of the arch shed pillar can be indirectly inferred through a change in the motor current of the electric push rod, to achieve real-time monitoring of the cuttage depth for an arch shed pillar.

## 4. Simulation Analysis of Cuttage Device

SOLIDWORKS was used to simulate and design the force feedback automatic cuttage device (Figure 6). The model included a walking chassis, a conveying platform, and an actuator of gantry structure. During operation, the arch shed pillar (4) was manually placed on the conveying platform (8) of the walking chassis, and the arch shed pillar (4) was then transported successively to the front of the machine by the conveying platform (8). On the conveying platform, two rows of hollow columns were evenly distributed with a height of 3 cm. The inner diameter of the hollow columns was 1.2 cm, which was slightly larger than the diameter of the arch shed pillars. The distance between the two rows of hollow columns was equal to the span of the arch shed pillars. The two lower end points of the arch shed pillars were placed in the hollow columns, which ensured that the pillars could be easily clamped by the execution end and transported to the tail of the fuselage while being attached on the conveying platform. Subsequently, the electric push rod (7) caused the execution end (5) to fall and clamp the arch shed pillar (4), causing it to rise. Then, the servo motor drove the lead screw sliding table (2) to transport the arch shed pillar (4) to the rear of the machine. After the transportation, the electric push rod (7) extended downward to press the arch shed (4) into the soil to achieve pillar cuttage, which performed repeatedly. In this operation, the execution end (5), the installation frame (6), and the electric push rod (7) were used to cut the arch shed pillar. Simultaneously, they were used to detect the cuttage depth of the arch shed pillar in real time.

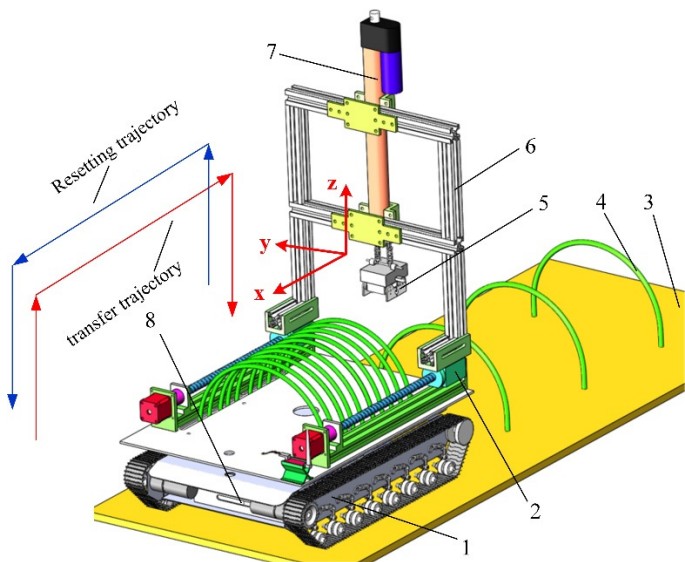

**Figure 6.** Simulation design diagram of the force feedback arch shed pillar cutting device. 1: walking chassis; 2: lead screw sliding table; 3: soil; 4: arch shed pillar; 5: execution end; 6: installation frame; 7: electric push rod; and 8: conveying platform.

### 4.1. Simulation Analysis of the Motion Trajectory of the Execution End

The execution end pulled the cuttage pillar from the front of the machine and transferred it to the rear, extending the conveying route and eliminating the vibration of the cuttage device that was generated by the short distance and the frequent starting and stopping of the motor. Therefore, the motion trajectory of the execution end was similar to the "door" shape in Figure 6, which can be further sorted into the black line through AB'C'G in Figure 7. The operation speed and the acceleration were discontinuous during the transportation of the pillar. Jumps in the displacement, speed, or acceleration caused mechanical vibration and, in severe cases, fractures in the machine material due to the right angle at the connection between the vertical and horizontal motion trajectory at the execution end. To make the motion of the execution end smooth and to eliminate the motion impact, the Láme curve [31,32] was used to optimize the operation trajectory of the execution end. The optimized trajectory is shown by the red curve through ABCDEFG in Figure 7, including the vertical rising segment AB, the horizontal moving segment CE, the vertical falling segment FG, and two transitional arcs $\overset{\frown}{BC}$ and $\overset{\frown}{EF}$.

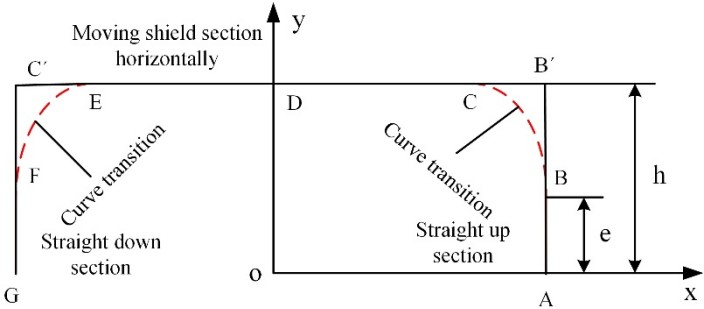

**Figure 7.** The motion trajectory of the execution end.

Given that the motion position point of the execution end is $P(t_i)$, and the distance traveled by the execution end is $s(t_i)$, the function for the relationship between the speed, the acceleration, and the displacement of the execution end with time t in the Cartesian coordinate system is as follows:

(1) When $0 \leq s(t_i) \leq h$, the execution end moves along the AB segment, and the kinematic parameters of the execution end are:

$$\begin{cases} P(t_i) = \begin{pmatrix} x_A & y_A & z_A + s(t_i) \end{pmatrix}^T \\ v(t_i) = \begin{pmatrix} 0 & 0 & s'(t_i) \end{pmatrix}^T \\ a(t_i) = \begin{pmatrix} 0 & 0 & s''(t_i) \end{pmatrix}^T \end{cases} \tag{7}$$

(2) When the execution end moves within the $\overset{\frown}{BC}$ segment along the Láme curve, the kinematic parameters of the execution end are solved according to the arc differentiation method, as shown in Figure 8:

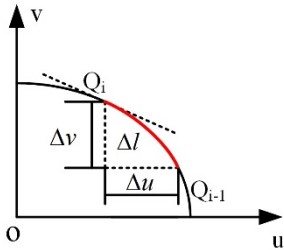

**Figure 8.** Schematic diagram of the Láme curve arc differential.

$$\Delta v = \frac{\Delta l}{\sqrt{1 + (u_i')^2}} \tag{8}$$

The relationship between point $Q_i$ and point $Q_{i-1}$ satisfies $v_i = v_{i-1} + \Delta v$, and is substituted into Equation (8):

$$u_i = g^n \sqrt{1 - \left(\frac{v_i}{f}\right)^n} \tag{9}$$

(3) When the execution end moves within the CD segment, the motion parameters of the execution end at time $t_i$ are:

$$\begin{cases} \Delta s_i = \Delta u_i \\ \Delta v_i = 0 \end{cases} \tag{10}$$

(4) The left and right sides of the motion trajectory of the execution end are symmetrical and the physical parameters of the second half of the motion are determined as described above.

From the above analysis, the curve of the distance s of the execution end versus time is shown in Figure 9. MATLAB was used to simulate and verify the motion of the execution end. The motion trajectory of the execution end required a smooth transition in all directions. According to the structure and parameters of the cuttage device, the obstacle avoidance height was $e$ = 5 cm, the value of the Láme curve was $g$ = 12 cm, $f$ = 2 cm, $m$ = 2, the horizontal transportation distance was 60 cm, and the one-way motion cycle was $T$ = 2 s. The motion trajectory of the execution end and the curves of displacement $s$, speed $v$ and acceleration $a$ in the three $xyz$ spaces versus time t in the Cartesian coordinate system are shown in Figure 10. The results in Figure 10 show that the curves of the displacement, the speed, and the acceleration of the execution end were smooth and continuous, with a zero value at the start and end points. Therefore, the trajectory optimization method can find the optimal motion law and prior dynamic performance of the execution end.

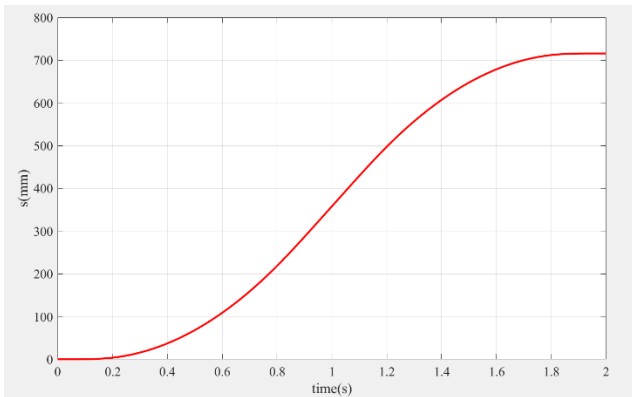

**Figure 9.** The distance curve of the execution end.

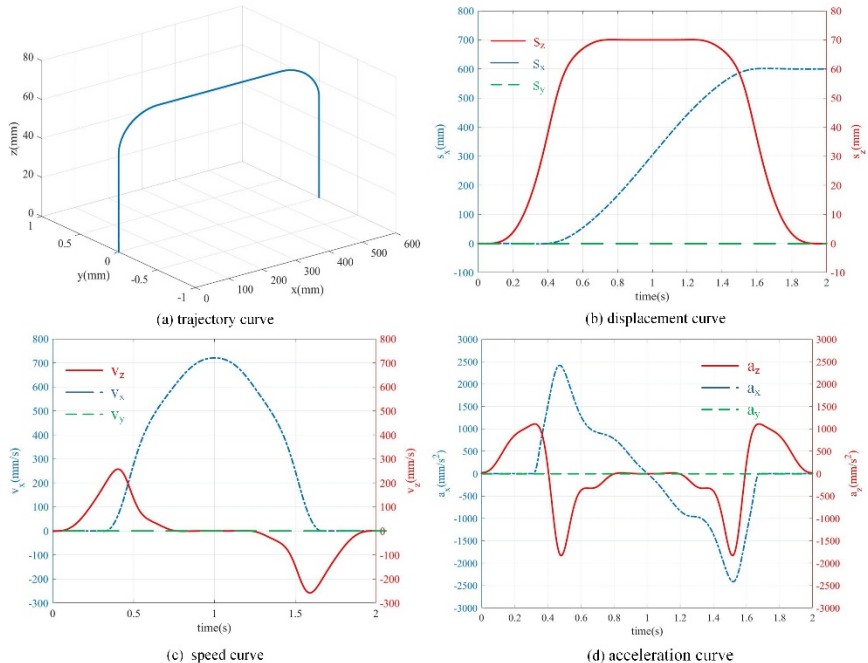

**Figure 10.** The motion parameter curves of the execution end.

### 4.2. Modal Analysis of the Installation Frame

The frequent starting and stopping of the execution end of the cuttage device caused alternating loads to act on the installation frame. When the vibration and impact frequency generated by an external force are close to the inherent frequency of the installation frame, resonance will occur, thus accelerating the failure of the installation frame and affecting its working reliability. The modal analysis method described in [33–37] must be used to obtain the modal parameters of the installation frame and study its vibration characteristics.

Let the matrices of the mass, damping, and stiffness of the system structure be *M*, *C*, and *K*, respectively. If *x* is the displacement vector of structural vibration, the external excitation of the system is *F* = 0, and the system damping is *C* = 0, then the system is underdamped without external force input, and its free vibration equation is:

$$M\ddot{x} + Kx = 0 \tag{11}$$

When the free vibration of the system is simple harmonic vibration, the vibration mode vector is $\phi$, the modal frequency is $\omega$, and the simple harmonic vibration equation is:

$$x = \phi e^{j\omega t} \tag{12}$$

Substituting Equation (12) into Equation (11) results in:

$$(K - \omega^2 M)\phi = 0 \tag{13}$$

If there is a non-zero solution to Equation (13), the expression of the inherent frequency and the vibration mode of the installation frame is:

$$\left| K - \omega^2 M \right| = 0 \tag{14}$$

Because low-order vibration has a considerable impact on the dynamic characteristics of the structure, and during the operation of the cuttage device in the field, the first four order frequencies of the installation frame were analyzed using SOLIDWORKS to obtain the inherent vibration frequency and vibration mode nephogram, as shown in Figure 11.

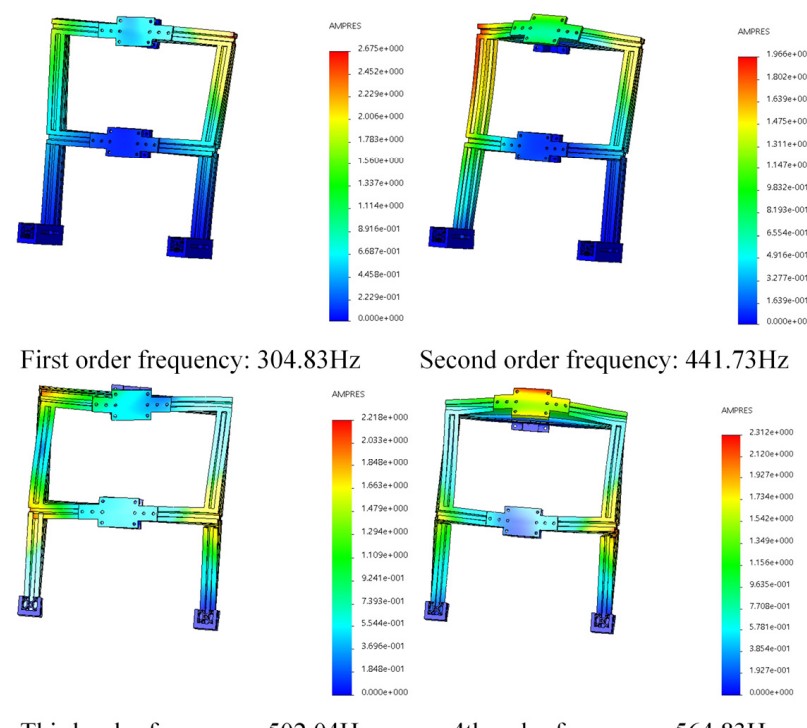

First order frequency: 304.83Hz     Second order frequency: 441.73Hz

Third order frequency: 502.04Hz     4th order frequency: 564.83Hz

**Figure 11.** The vibration nephogram.

## 5. Experimental Results and Analysis

### 5.1. Construction of Entity System

An automatic cuttage device for an arch shed pillar was built based on verification using the simulation results (Figure 12). The installation frame of the cuttage device was an aluminum profile, and the servo motor (Time Chaoqun Technology Co., Ltd., Beijing, China) was used to drive the roller lead screw (800 mm, FY05, Louis Martin Flagship Store, Yancheng, China) and transport the pillar. The pillar cuttage was performed using a DC brushless electric push rod (350 mm, Hansen Motor Tools Co., Ltd., Wenzhou, China) with a 350 mm stroke, and a steering gear (DS3218, Premium Robot Co., Ltd., Shenzhen, China) was used to drive the execution end to grasp the pillar. Other non-standard components were 3D printed with PLA and photosensitive resin.

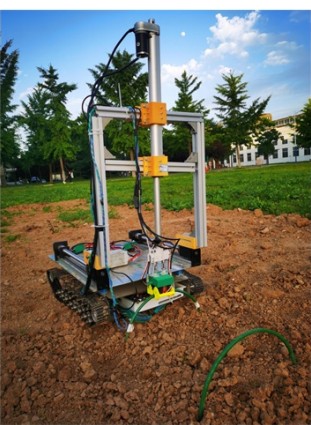

**Figure 12.** The automatic cuttage device for an arch shed pillar.

The control system of the cuttage device comprised a main control board (STM32F103VET6, Guangzhou Xingyi Electronic Technology Co., Ltd., Guangzhou, China), a servo motor driver (DM-055B, Time Chaoqun Technology Co., Ltd., Beijing, China), a brushless motor driver (AQMD360BLS, Love Control Electronics·Aisikong Co., Ltd., Chengdu, China) and a steering gear driver (LSC-16-V1.3, Magic Robot Official Mall, Shenzhen, China). Based on the control system, a cuttage depth sensing system for an arch shed pillar with "current–force–depth" real-time feedback was constructed, as shown in Figure 13. The main controller collected the current information of the electric push rod motor through the motor driver at a frequency of 20 Hz, detected the rotation position of the motor through the Hall encoder, and received the command sent by the main controller to drive the motor to perform the corresponding actions.

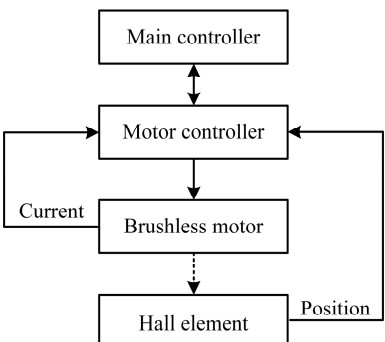

**Figure 13.** The cuttage depth sensing system of the arch shed pillar with "current–force–depth" real-time feedback.

The "current–force–depth" feedback control process is shown in Figure 14. During the operation, the motor controller was set to work mode in the closed-loop speed mode and operated at a constant speed to equalize the electromagnetic torque of the motor and the load torque. After data processing, the commutation pulse number and phase current information of the motor were read to obtain the phase current value and the rotation position of the electric push rod motor. After filtering, the motor phase current value $I_t$ was obtained under different cuttage depths. A current jumping threshold was then specified as $\varepsilon$. When the actual current $I_r$ and theoretical current $I_t$ values during the operation satisfied $|I_t - I_r| > \varepsilon$, the motor braked and the electric push rod ceased to function.

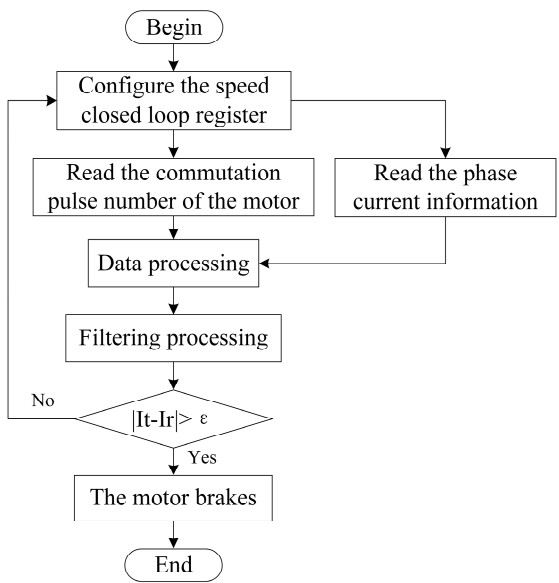

**Figure 14.** The "current–force–depth" feedback control process.

*5.2. Verification Experiment of the Cuttage Depth*

5.2.1. Verification Experiment of "Current Depth" Real-Time Feedback

The real-time current of the motor at different cuttage depths was measured under three operation conditions (Table 1) to assess the feasibility of the control system. The cuttage speed was set to 28 mm/s, and the real-time current of the motor was read at different cuttage depths. A scatter plot of the changes in the motor current is shown in Figure 15, where it is clear that when the cuttage depth was 10 cm, the current values of the motor were 230 mA, 150 mA, and 60 mA for cuttage in sandy soil, dry soil, and wet soil, respectively. At the same time, with an increase in the cuttage depth, the motor current also increased and the cuttage depth information could be obtained through the current.

**Table 1.** Physical parameters of the experimental soil.

| Soil Type | Moisture Content (%) | Soil Firmness at Different Depths (kPa) | | | | |
|---|---|---|---|---|---|---|
| | | 2.5 cm | 5 cm | 7.5 cm | 10 cm | 12.5 cm |
| Sandy soil | 3.25 | 291.00 | 747.71 | 1604.43 | 1818.14 | 2141.29 |
| Dry soil | 19.37 | 162.17 | 544.33 | 720.17 | 1141.50 | 1800.17 |
| Wet soil | 37.84 | 159.20 | 427.40 | 606.20 | 622.80 | 725.00 |

5.2.2. Determination Experiment of the Cuttage Speed

The operation efficiency increases when cuttage is performed at a faster speed. Accordingly, a greater instantaneous impact force will act on the pillar, and the current value will change. To determine the most suitable cuttage speed, the real-time current was measured with a selected cuttage depth of 10 cm and a cuttage speed that varied in the range of 10–28 mm/s. The experimental data, which are presented as a scatter plot of the current in Figure 16, show that a change in cuttage speed does not cause a change in the current value in a wet soil environment. The motor current showed a rising trend with an increase in cuttage speed in the sandy soil environment, while in the dry soil environment, the motor current increased at first and then plateaued. Comprehensive consideration of the operation efficiency and safety of the motor suggests that the cuttage speed should be appropriately increased to improve the cuttage efficiency in wet and dry soil environments. When cutting in a sandy soil environment, the cuttage speed should be appropriately reduced to decrease the current and to protect the motor.

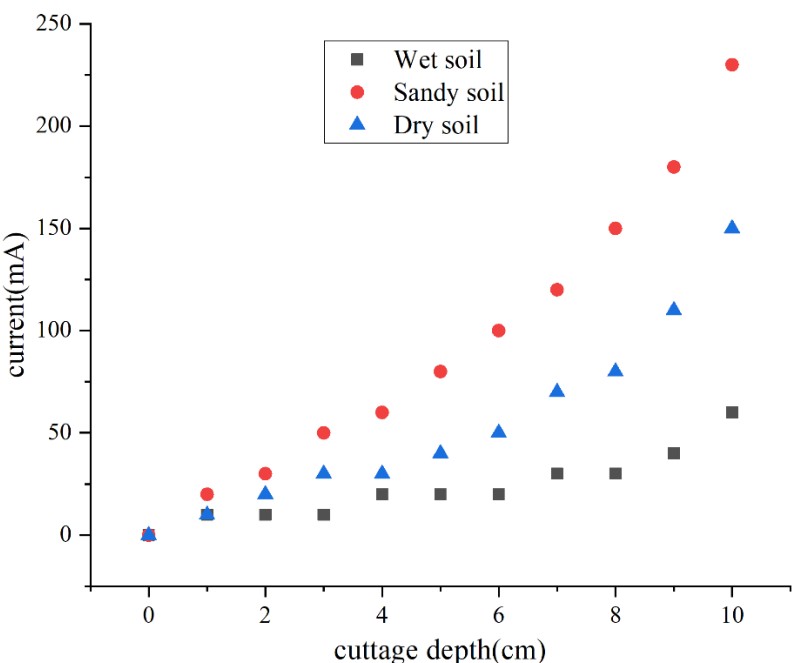

**Figure 15.** The verification experiment of the "current depth" real-time feedback.

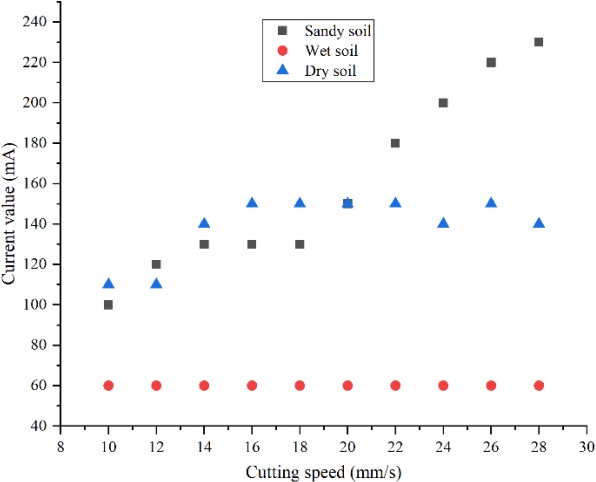

**Figure 16.** The determination experiment of cuttage speed.

### 5.2.3. Error Calibration of the Cuttage Depth

As mentioned above, the optimal stability of the arch shed was at cuttage depth of 10 cm. The motor was controlled for cuttage at different speeds, and the errors during cuttage in sandy soil (a), dry soil (b), and wet soil (c) with critical values of 230 mA, 150 mA, and 60 mA, respectively, were recorded. The experiment was repeated 40 times and an error bar for cuttage depth was drawn, as shown in Figure 17.

The results shown in Figure 17 indicate that the errors in cuttage depth were within 7 mm when cutting in dry and sandy soil environments, and within 5 mm when cutting in a wet soil environment. The error remained under 5 mm with an increase in cuttage speed. Therefore, the device was suitable for cuttage operations under wet soil conditions, which is favorable for controlling the amount of cuttage error and improving the operation efficiency.

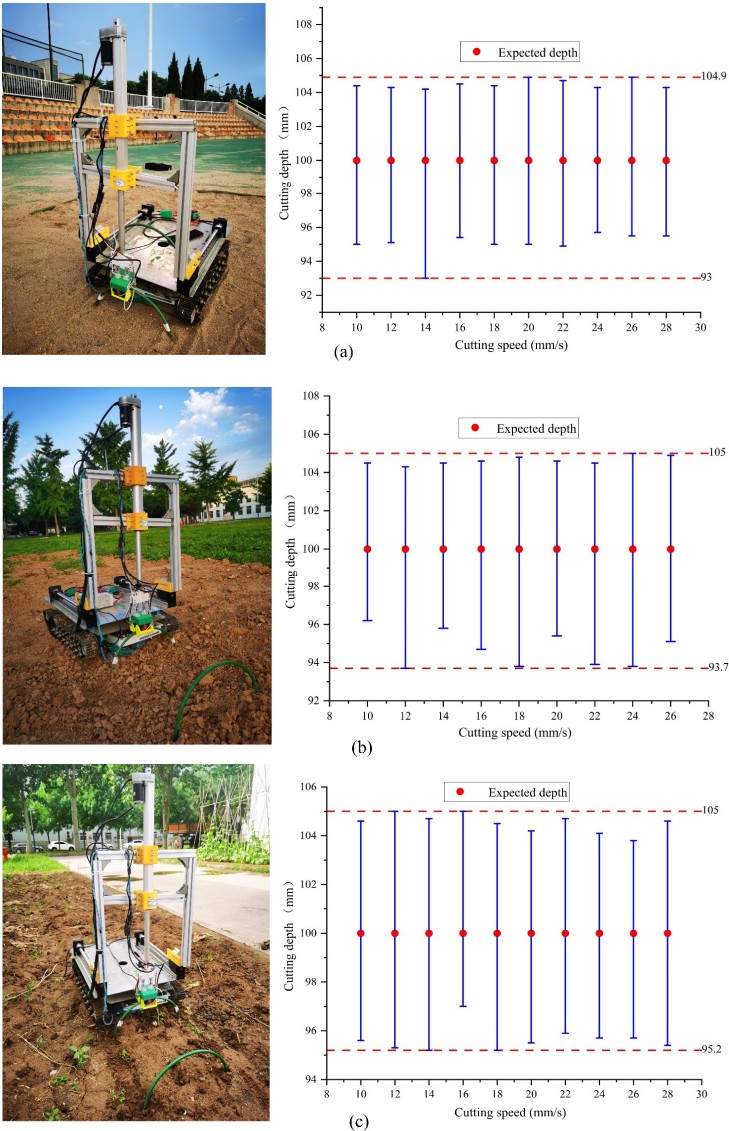

**Figure 17.** Error bar charts for the cuttage depth. Note that (**a**): Sandy soil experiments; (**b**): dry soil experiments; (**c**): wet soil experiments.

### 5.3. Experiment of Trajectory Planning

To assess the feasibility of trajectory planning, an experiment was performed at the execution end of the cuttage device. The specific experimental process was as follows:

(1) According to the transportation and distance lifting of the pillar, the trajectory of the execution end was planned to obtain the theoretical data, and the theoretical curve was generated using MATLAB;

(2) The motion trajectory of the execution end was tracked and measured, the experiment was repeated five times, and the collected data was used to generate the actual trajectory route;

(3) The coincidence degree between the theoretical calculated trajectory and the actual trajectory was compared, as shown in Figure 18.

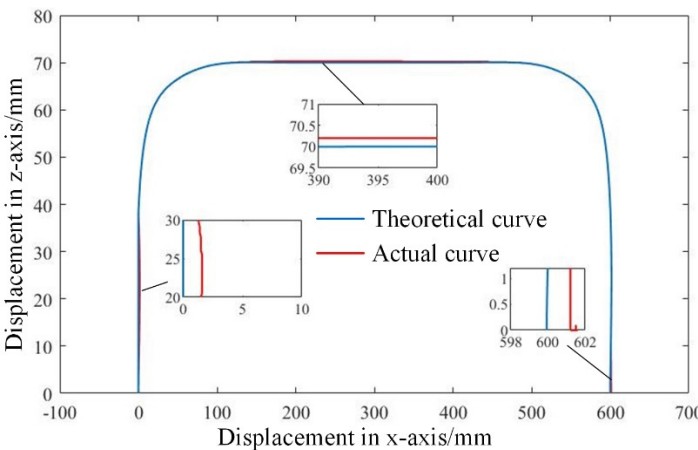

**Figure 18.** The verification test of the trajectory planning at the execution end.

In the experiment, the transportation and lifting actions of the pillar were achieved based on the theoretical calculated trajectory for the execution end in a continuous and relatively stable action. The comparison of the actual trajectory with the theoretical trajectory showed that they essentially coincided, with a few errors. The errors were concentrated in the initial stage and the foothold, and the maximum error was 2.1 mm, which was within the allowable error range and met the design requirements.

### 5.4. Modal Experiment Analysis

To verify the results of the modal simulation analysis, the modal characteristics of the installation frame were tested by using the pulse excitation method, and the experimental modal test and analysis system was built, as shown in Figure 19. The analysis system was composed of a modal force hammer (LC02, Donghua Testing Technology Co., Ltd., Taizhou, China), an acceleration sensor (DH186, Donghua Testing Technology Co., Ltd., Taizhou, China), and a data acquisition and analysis module (DH5920, Donghua Testing Technology Co., Ltd., Taizhou, China).

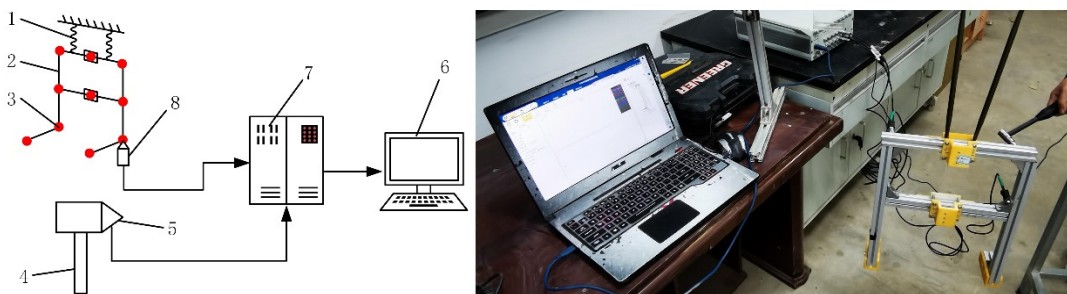

**Figure 19.** Experimental modal test and analysis system. 1: spring; 2: installation frame; 3: test point; 4: modal force hammer; 5: force sensor; 6: PC; 7: data acquisition and analysis module; and 8: acceleration sensor.

During the experiment, two springs were used to suspend the installation frame in a static state in the air. An acceleration sensor was arranged at the cross-linking point of the aluminum profile of the installation frame. The excitation point of the mounting frame was knocked hard along the longitudinal direction and the signal was transmitted to the data acquisition and analysis module. After being processed by the dynamic signal acquisition and analysis system, DHDAS 6.8.18, the modal parameters of the installation frame were obtained (Table 2). The results of the modal finite element analysis (shown in Table 2) were essentially consistent with those of the modal test analysis for the installation frame. The first four modal inherent frequencies of the installation frame were concentrated in the

303–565 Hz range with 2.32% of the maximum inherent frequency error, indicating the reliability of the modal finite element analysis. During the transportation of the cuttage pillar, the power was provided by the motor. The maximum speed of the motor output shaft was 3000 r/min, suggesting a frequency of 50 Hz, which was far lower than 296.58 Hz. Therefore, the random vibration with a frequency lower than 50 Hz was transmitted to the installation frame, resulting in a small possibility of resonance.

**Table 2.** Comparison of modal results between the finite element analysis and the test.

| Order | Frequency | | Vibration Shape | | Frequency of Error |
|---|---|---|---|---|---|
| | Modal Finite Element Analysis (Hz) | Modal Test Analysis (Hz) | Modal Finite Element Analysis | Modal Test Analysis | |
| 1 | 303.47 | 296.58 | bending | bending | 2.32% |
| 2 | 441.18 | 443.94 | bending | bending | −0.62% |
| 3 | 500.69 | 506.72 | bending + torsion | bending + torsion | −1.19% |
| 4 | 564.62 | 554.53 | bending + torsion | bending + torsion | 1.82% |

Note: frequency of error = (modal finite element analysis − modal test analysis)/modal test analysis.

## 6. Conclusions

In this study, an automatic cuttage device for an arch shed pillar based on force feedback was proposed, which provides a new idea for the automatic cutting of arch shed pillars.

(1) The wind resistance of a small arch shed was analyzed. The simulation analysis by ANSYS software showed that the wind resistance was better when the cutting depth was 10 cm.

(2) Three-dimensional modeling of the cuttage device was carried out through SOLID-WORKS software, the trajectory of the execution end was planned, and the relevant motion parameters were solved through MATLAB software simulation to avoid motion impact and make the motion trajectory smoother.

(3) A modal analysis and experimental verification were carried out for the installation frame. The experiment showed that the resonance frequency was in the range of 303–565 Hz, and the device did not resonate during the operation.

(4) A real-time feedback mechanism based on "current–force–depth" was built, which could feed back the current of the electric pusher motor in real-time. Based on this, the cutting depth was estimated in real time, allowing for the convenient building of an equal-depth pillar cuttage device. Even under the condition of the stress limit of the pillar, the automatic cutting device was able to realize "overload stop" without causing damage to the pillar, and realize the overload protection of the pillar. This could have a positive role in promoting the technical development and social application of automatic arch shed construction.

Due to the limitations of time and funds, this study did not investigate the interaction mechanisms of "arch shed pillar soil", which have certain limitations. In the future, this further study will be carried out, as well as a study on the linear cutting of arch shed pillars, to further explore the operation efficiency in the process of automatic cutting of arch shed pillars.

**Author Contributions:** Conceptualization, K.C.; methodology, X.L.; software, S.J.; validation, X.H.; formal analysis, L.L.; investigation, T.W.; resources, G.M.; data curation, K.C.; writing—original draft preparation, K.C.; writing—review and editing, W.L.; visualization, Y.S.; supervision, W.L.; project administration, W.L.; funding acquisition, W.L. All authors have read and agreed to the published version of the manuscript.

**Funding:** This research was funded by the Wang Tongchuan Innovation and Entrepreneurship Fund Project of Northwest A&F University (4015401515010118), the Key Research and Development Program—Major (Key) Projects of the Ningxia Hui Autonomous Region (2019BFF02003) and the Key Industry Chain Innovation Project of the Shaanxi Province (2018ZDCXL-NY-03-06).

**Institutional Review Board Statement:** Not applicable.

**Informed Consent Statement:** Not applicable.

**Data Availability Statement:** Not applicable.

**Conflicts of Interest:** The authors declare no conflict of interest.

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
