# Peer review of "Design of and Experiments with an Automatic Cuttage Device for an Arch Shed Pillar with Force Feedback"

_agriculture, doi:10.3390/agriculture12060875_

Round 1

Reviewer 1 Report

It is a very interesting topic, but I have comments on this article.

The purpose of the work has not been clearly defined. Indicate clearly the purpose of the work.

The conclusions are not consistent with the introduction presented and should be consistent with the purpose of the study.

The presented work is too limited in the bibliography. Please, enrich your introduction.

  I hope my comments enrich your work.

Author Response

Response to Reviewer 1 Comments

Dear Reviewer:

Thank you for your comments concerning our manuscript entitled "Design and Experiment of Automatic Cuttage Device of an Arch Shed Pillar with Force Feedback" (ID: 1745113).". Those comments are all valuable and very helpful for revising and improving our paper, as well as the important guiding significance to our researches. We have studied comments carefully and have made correction which we hope meet with approval. Revised portion are marked in yellow in the paper. The main corrections in the paper and the responds to the reviewer’s comments are as flowing:

Responds to the reviewer’s comments:

Point 1: The purpose of the work has not been clearly defined. Indicate clearly the purpose of the work.

Response: Thank you very much for your comment. And I’m sorry for making such a mistake. After investigation and analysis of the current situation of arch shed construction, we want to make an automatic cutting device for arch shed pillars with pillar protection function. This is the main purpose of our work. At the same time, in order to ensure the good performance of the automatic cutting device for arch shed pillars, we have also done some other work, such as: modal analysis, execution of end trajectory planning, etc. Of course, all the work is carried out around the realization of the automatic cutting operation of the arch shed pillars. We have made changes in the Abstract and Introduction, which we hope will improve our paper. The modified part has been marked in yellow in the text. (Line 11 and 51 in the text)

Point 2: The conclusions are not consistent with the introduction presented and should be consistent with the purpose of the study.

Response: Thank you very much for your comment. And according to your opinion, we have revised the conclusion. The modified conclusion is as follows:

In this study, the automatic cutting device of arch shed pillar based on force feed-back is proposed, which provides a new idea for the automatic cutting of arch shed pillar.

(1) The wind resistance of the small arch shed is analyzed. The simulation analysis by ANSYS software shows that the wind resistance is better when the cutting depth is 10cm.

(2) Carry out three-dimensional modeling of the cutting device through SolidWorks software, plan the trajectory of the execution end, and solve the relevant motion pa-rameters through MATLAB software simulation to avoid motion impact and make the motion trajectory smoother.

(3) Modal analysis and experimental verification are carried out for the installation frame. The experiment shows that the resonance frequency is in the range of 303~565hz, and the device will not resonate during operation.

(4) A real-time feedback mechanism based on "current force depth" is built, which can feed back the current of the electric pusher motor in real time. Based on this, the cutting depth can be estimated in real time, and it is convenient to build a pillar equal depth cutting device. Even under the condition of the stress limit of the pillar, the automatic cutting device can realize "overload stop" without worrying about the damage of the pillar, and realize the overload protection of the pillar, It has a positive role in promoting the technical development and social application of automatic arch shed construction.

Due to the limitation of time and funds, this study did not study the interaction mechanism of "arch shed pillar soil", which has certain limitations. In the future, this study will be further carried out, as well as the linear cutting of arch shed pillar, to further explore the operation efficiency in the process of automatic cutting of arch shed pillar.

The modified part has been marked in yellow in the text.

Point 3: The presented work is too limited in the bibliography. Please, enrich your introduction.

Response: Thank you very much for your comment. During the initial research, we consulted a large amount of literature. Unfortunately, there are indeed fewer papers on the construction machinery of arch sheds. Based on your comments, we have revised the Introduction again and introduced a new reference, which has been inserted in the text. At the same time, we have also clarified the purpose of this research, which has been revised and added to the text, hoping to help improve the quality of the paper. Finally, all revisions are marked in yellow in the text. Expect to be able to meet the journal's requirements.

Special thanks to you for your good comments.

Reviewer 2 Report

Although the text looks good in general, but I think that the word 'cut' in the title and at many points in the text is not appropriate because it refers to the pieces penerate in the soil, instead the 'cut' word 'penetration' should be usedin the whole text .

Author Response

Response to Reviewer 2 Comments

Dear Reviewer:

Thank you for your comments concerning our manuscript entitled "Design and Experiment of Automatic Cuttage Device of an Arch Shed Pillar with Force Feedback" (ID: 1745113).". Those comments are all valuable and very helpful for revising and improving our paper, as well as the important guiding significance to our researches. We have studied comments carefully and have made correction which we hope meet with approval. Revised portion are marked in yellow in the paper. The main corrections in the paper and the responds to the reviewer’s comments are as flowing:

Responds to the reviewer’s comments:

Point: I think that the word 'cut' in the title and at many points in the text is not appropriate because it refers to the pieces penetrate in the soil, instead the 'cut' word 'penetration' should be used in the whole text.

Response: Dear reviewer, thank you very much for your contribution in the review process of this paper, and thank you very much for your valuable comments. In the text, the word "cuttage" means the pillar was inserted into the soil. The process is similar to that of plant cutting propagation, so we used the word "cuttage" instead of "penetration". At the same time, we refer to the title and content of the article entitled "Design and experiment of single-row double cuttage and film covering multi-functional machine for low tunnels" and the paper “Effects of period of cuttage on growth of seedling for Lonicera japonica”, feeling that "cuttage" can better describe the fact that the pillars are inserted into the soil process, so we finally chose "cuttage".

Special thanks to you for your good comments.

Reviewer 3 Report

In my opinion the presented solution is not the best idea. According to the paper it works, but the machine is big, slow and complicated. It doesn't have leveling system. It grabs arches in one point and it deflects it during pushing.  

Some remarks:
- units should be converted to SI system (cm > mm/m),
- Pa MPa - there is "a" as a subscript it should be normal size letter,
- generally strain has no unit,
- there are different fonts, size, some figure captions start with a small letter,  etc - the article has to be unified,
- Fig 15 - it would be better to give force or torque instead of current,
- explain how arches are "attached" to the platform of the machine,

Author Response

Response to Reviewer 3 Comments

Dear Reviewer:

Thank you for your comments concerning our manuscript entitled "Design and Experiment of Automatic Cuttage Device of an Arch Shed Pillar with Force Feedback" (ID: 1745113).". Those comments are all valuable and very helpful for revising and improving our paper, as well as the important guiding significance to our researches. We have studied comments carefully and have made correction which we hope meet with approval. Revised portion are marked in yellow in the paper. The main corrections in the paper and the responds to the reviewer’s comments are as flowing:

Responds to the reviewer’s comments:

Point 1: units should be converted to SI system (cm > mm/m);

Response: I'm sorry we didn't notice such details. According to your opinion, we have replaced all units with SI system. The modified part has been marked in red in the text. When measuring the error, we chose the vernier caliper as the measuring tool, so the last reserved unit is mm. When introducing the material, we attached the model of the material, which can be seen on the material nameplate, so the unit we reserve is also mm. We don't know if this is compliant and look forward to hearing from you.

Point 2: Pa MPa - there is "a" as a subscript it should be normal size letter;

Response: I'm sorry we didn't notice such details. According to your opinion, We have replaced "a" in all units "pa", "MPa" with normal size. The modified part has been marked in red in the text. (Lines 180.181.204 and 205 in the text)

Point 3: generally strain has no unit,

Response: We are sorry that we made such a mistake. In the simulation analysis, the unit of strain is shown in the simulation result graph, so we also add it in the paper. After your reminder, we have modified the content. Besides, the modified part has been marked in red in the text. (Lines 202 and 203 in the text)

Point 4: there are different fonts, size, some figure captions start with a small letter, etc - the article has to be unified;

Response: We are sorry that we made such a mistake. And we have reviewed and revised the text format.

Point 5: Fig 15 - it would be better to give force or torque instead of current,

Response: Thank you very much for your review. In this article, we use the cutting resistance as the intermediary to estimate the cutting depth, and finally achieve the purpose of equal depth cutting, but the final form is current, so the final data we measured in Figure 15 is current. At the same time, in order to realize equal depth cutting, we must use the current value under different depths to control whether the electric pusher motor rotates, so as to achieve the purpose of unequal depth cutting. Therefore, in this process, the current value has become the key physical quantity to realize equal depth cutting. So what we finally present in Figure 15 is the current value rather than the torque value.

Point 6: explain how arches are "attached" to the platform of the machine,

Response: Thank you very much for your comment. First of all, when working, we manually place all the pillars on the conveying platform in advance. On the conveying platform, we evenly distributed two rows of hollow columns with a height of 3cm. The inner diameter of the hollow columns is 1.2cm, which is slightly larger than the diameter of the pillars of the arch shed. The distance between the two rows of hollow columns is equal to the span of the pillars of the arch shed. When placing the pillars, we place the two lower end points of the arch shed pillars in the hollow columns, which can ensure that the pillars can be easily clamped by the execution end and transported to the tail of the fuselage while being attached on the conveying platform. After your reminder, we have modified the content. Besides, the modified part has been marked in red in the text. (Line 271 in the text)

Special thanks to you for your good comments.